# Relationship between Individual Social Capital and Cognitive Function among Older Adults by Gender: A Cross-Sectional Study

**DOI:** 10.3390/ijerph16122142

**Published:** 2019-06-17

**Authors:** Tomoko Ito, Kenta Okuyama, Takafumi Abe, Miwako Takeda, Tsuyoshi Hamano, Kunihiko Nakano, Toru Nabika

**Affiliations:** 1Department of Nursing, Faculty of Nursing and Nutrition, The University Shimane, 151 Nishihayashigi-cho, Izumo, Shimane 693-8550, Japan; 2Center for Community-Based Healthcare Research and Education (CoHRE), Organization for Research and Academic Information, Shimane University, 223-8 Enya-cho, Izumo-shi, Shimane 693-8501, Japan; oku117@med.shimane-u.ac.jp (K.O.); t-abe@med.shimane-u.ac.jp (T.A.); cohre1@med.shimane-u.ac.jp (M.T.); k-nakano@riko.shimane-u.ac.jp (K.N.); nabika@med.shimane-u.ac.jp (T.N.); 3Department of Sports Sociology and Health Sciences, Faculty of Sociology, Kyoto Sangyo University, Motoyama, Kamigamo, Kita-ku, Kyoto 603-8555, Japan; thamano@cc.kyoto-su.ac.jp; 4Department of Functional Pathology, School of Medicine, Shimane University, 89-1 Enya-cho, Izumo, Shimane 693-8501, Japan

**Keywords:** cognitive function, social capital, gender, rural areas, dementia prevention, adult men, adult women, elderly

## Abstract

As it is not easy to modify lifestyle, it is important to examine the effect of social capital (SC), which does not require behavior modifications, on dementia prevention. This study aimed to clarify gender differences in the relationship between cognitive function and individual SC among people living in a rural area in Japan. We used the Shimane Center for Community-based Healthcare Research and Education (CoHRE) study data from 2011 to conduct a cross-sectional analysis. The analysis included 491 participants, aged 40 years or older, who had undergone medical examinations in two rural towns in Japan. Both cognitive SC and structural SC were measured. Multivariate logistic regression analysis was conducted to estimate the odds ratios (OR) and 95% confidence interval (CI) for cognitive function levels as binary outcomes. We found a significant association between cognitive function and individual cognitive SC in men (OR: 3.11, 95% CI: 1.43–6.78), and found that cognitive function was associated with structural SC in women (OR: 1.89, 95% CI: 1.08–3.31). This study showed that the relationship between cognitive function and individual SC differed by gender. These results suggest that it is important to approach dementia prevention differently in men and women.

## 1. Introduction

Dementia is a state of impaired social functioning marked by memory disorders and impaired judgment due to an organic brain disorder. The number of people with dementia worldwide is increasing, reaching 47 million in 2015. Furthermore, the number of people with dementia is projected to increase to about 66 million in 2030 and 131 million in 2050 [1], making dementia prevention an urgent global issue. Multiple studies have shown that hypertension, diabetes, obesity, dyslipidemia, smoking, a lack of physical activity, and other factors are suspected lifestyle risk factors that affect cognitive function [2,3,4]. However, it is not easy for elderly individuals to modify their lifestyles, and behavior modification can result in a decreased quality of life [5,6].

Recently, social capital (SC) has attracted attention as a means to improve cognitive function [7,8]. Social capital is defined as the features of social groups, such as trust, norms, and networks, that improve social functioning by facilitating coordinated actions of the constituents [9]. Measurement of SC in public health often adopts a social cohesiveness approach. Since 1990, many studies have elucidated the relationship between SC and health. For example, Kawachi et al. [10] found the association between social trust and total mortality in the United States. Hamano et al. [11] and Nakamine et al. [12] found a relationship between SC and mental health in Japan, and Ferlander et al. [13] reported the relationship between SC and self-perceived health in Russia. In Asia, Chuang et al. [14] observed a relationship between SC and drinking and smoking as lifestyle habits in Taiwan. Hamano et al. [11] and Nakamine et al. [12] found a relationship between SC and mental health in Japan. For the studies of cognitive function in Asia, Wang et al. [7] found a relationship between SC and mild cognitive impairment in China. Sakamoto et al. [8] reported a relationship between cognitive function and social participation in community-dwelling older populations in Japan.

For cognitive function, the differences by gender have been clarified [15,16]. However, differences by gender for SC have not been fully discussed with respect to the association between SC and health. A recent study reported that differences by gender existed in social networks [17]. Moreover, a previous study reported that SC was not an equally effective resource for everyone, and it had adverse effects on social equality between men and women [18]. Therefore, a better understanding of the function of SC is required to capture the association between SC and cognitive function by gender among rural older adults. Therefore we hypothesized that there may be gender differences in the relationship between cognitive function and SC.

The aim of this study was to assess the relationship between individual cognitive and structured SC, and cognitive function among community-dwelling Japanese older adults by gender.

## 2. Materials and Methods

### 2.1. Study Design

The study had a cross-sectional study design, and used the data from the Shimane Center for Community-based Healthcare Research and Education (CoHRE) study. The analysis included data from a total of 491 consecutively-enrolled participants (187 men and 304 women) who were recruited among people who had undergone a health examination in Ohnan-cho and Okinoshima-cho, two rural towns in the Shimane Prefecture of Japan, during 2011. The age of the participants ranged from 40 to 88 years. The study protocol was approved by the Research Ethics Review Committee of University of Shimane Prefecture (Protocol #254) on 9 August 2018, and the Ethics Committee of Shimane University on 9 May 2018, (Protocol #3149).

### 2.2. Measurement of Cognitive Function

Cognitive function was evaluated using the iPad version of the Cognitive Assessment for Dementia (CADi) [19]. The CADi evaluation tool measures cognitive function of a 10-point scale, in which a score of 7 is classified as mild cognitive impairment (MCI). For the purpose of analysis, we categorized cognitive function into two groups, based on the participants’ CADi scores. Participants with a CADi score of 0–7 were assigned to the low cognitive function group, and those with a CADi score of 8–10 were assigned to the high cognitive function group.

### 2.3. Measurement of Social Capital

This study measured two types of SC, namely cognitive SC and structural SC, both of which were measured by adopting the Harpham method [20]. Cognitive SC is the individual’s subjective perception about the cohesive characteristics of their belonging to a group (neighborhood), and structural SC refers to the individual’s participation behavior in formal and informal social groups.

In order to decide on a measure of cognitive SC, we consulted previous studies [11,20,21]. We based the measure of cognitive SC on participants’ responses to the question: “Would you say that people in your neighborhood can be trusted or that you need to be very careful in dealing with them?” Their answers were rated on a 10-point scale, with 1 indicating excellent trust, 9 indicating very low trust, and 10 being “do not know”. After excluding the participants who had answered “do not know” (*n* = 24), we categorized cognitive SC into two groups based on their answers to the question [11]. Participants with a value of 1–4 were assigned to the high trust category, and those with a value of 5–9 were assigned to the low trust category.

In order to decide on a measure of structural SC, we also consulted previous studies [11,20,21]. We assessed structural SC based on the number of different types of social groups to which participants belonged. Participants were asked whether they belonged to each of six different types of social groups: (1) local groups; (2) sports, recreation, hobby, or cultural groups; (3) alumni associations; (4) political organizations; (5) citizen groups; and (6) agriculture-related organizations. The respondents answered either “yes” or “no” to each question in order to calculate a structural SC score. A value of 1 was assigned if the answer was “yes”, and a value of 0 was assigned if the answer was “no”. Subsequently, an individual structural SC score, with a scale of 0–6, was calculated by summing up the values of the six social group participation variables. The scores were dichotomized, at the median, into two categories: <3 (reference) and ≥3.

### 2.4. Other Measures

Information of sex, age, regular exercise, smoking, and years of education was obtained using a questionnaire. Age was categorized into three groups: <65, 65–74, and ≥75 years. Body mass index (BMI) was calculated from height and weight, which were measured during medical examinations. We used Asian BMI reference cutoff values to categorize participants into following four groups: <18.5 (underweight), 18.5–22.9 (normal weight), 23.0–27.4 (overweight), and ≥27.5 (obese) [22]. The question about regular exercise was based on a question used in the National Health and Nutrition Survey, performed by the Japanese Ministry of Health, Labor and Welfare. The participants were asked: “Are you conducting exercise that makes you sweat, for more than 30 minutes a day, at least twice a week?” Regular exercise was categorized into two groups: no (reference) and yes. Depressive symptoms were assessed using the Zung Self-Rating Depression Scale (SDS), which consists of a 20-item self-reported questionnaire [23]. Each item is scored from 1 to 4, the total score ranges from 20 to 80, with higher scores indicating more severe depressive symptoms. A Japanese version of the SDS has been developed [24]. The SDS scores were categorized into two groups: <40 (reference) and ≥40 (having depressive symptoms). The participants were characterized as non-smokers (reference) or smokers. Smokers were defined as those who smoked at least one cigarette/day. Years of education was divided into two categories: <12 years (reference) and ≥12 years.

### 2.5. Statistical Analysis

The relationship between the cognitive function category and the other variables was analyzed using Pearson’s chi-squared test for the categorical variables. To investigate the relationship between cognitive function and the other variables, we performed multivariate logistic regression analyses using cognitive function as the dependent variable, and stratified the analyses by gender. We developed three models: Model 1 included structural SC without cognitive SC; Model 2 included cognitive SC without structural SC; and Model 3 included both structural and cognitive SC. All three models included age, BMI, regular exercise, depressive symptoms, smoking, and years of education as covariates. All statistical analyses were performed using SPSS statistical software (version 25, IBM Corporation, Tokyo, Japan), and *p*-values < 0.05 were considered to be statistically significant.

## 3. Results

### 3.1. Participants Characteristics

The characteristics of the study participants are shown in Table 1. The study included 187 men (mean age: 69.2 years) and 304 women (mean age: 68.9 years), with an age range of 40–88 years. Based on their CADi scores, 59.4% of the men and 57.2% of the women were in the higher cognitive function group.

In men, significant positive relationships with cognitive function were shown in high cognitive SC (low cognitive function group 50.7%, high cognitive function group 69.5%, *p* = 0.01) and years of education (low cognitive function group 40.8%, high cognitive function group 72.2%, *p* < 0.001). Significant negative relationships with cognitive function were shown in age (age < 65 low cognitive function group 15.8%, high cognitive function group 28.8%; age 65–74 low cognitive function group 64.5%, high cognitive function group 67.6%; age ≥ 75 low cognitive function group 19.7%, high cognitive function group 3.6%, *p* = 0.001) and depressive symptoms (low cognitive function group 47.3%, high cognitive function group 31.8%, *p* = 0.03).

In women, significant positive relationships with cognitive function were shown in structural SC (low cognitive function group 29.8%, high cognitive function group 48.8%, *p* = 0.001) and the years of education (low cognitive function group 32.0%, high cognitive function group 68.0%, *p* < 0.001).

Significant negative relationship with the cognitive function were shown in age (age < 65 low cognitive function group 13.1%, high cognitive function group 35.6%; age 65–74 low cognitive function group 65.4%, high cognitive function group 55.7%; age ≥ 75 low cognitive function group 21.5%, high cognitive function group 8.6%, *p* < 0.001), and depressive symptoms (low cognitive function group 45.7%, high cognitive function group 32.9 %, *p* = 0.02).

### 3.2. Results of the Multivariate Logistic Regtrssion Analysis

The results of the multivariate logistic regression analyses of the relationship between SC and cognitive function are shown in Table 2. In men, in Model 1 significant positive relationships with cognitive function variable were observed for years of education (odds ratio, OR: 2.84, 95% confidence interval, 95% CI: 1.37–5.91), and significant negative relationships with cognitive function were observed for age over 75 (OR: 0.19, 95% CI: 0.05–0.78, vs. <65 years) and depressive symptoms (OR: 0.42, 95% CI: 0.21–0.86). In Model 2, significant positive relationships with cognitive function were observed for cognitive SC (OR: 2.99, 95% CI: 1.42–6.33) and the years of education (OR: 3.05, 95% CI: 1.45–6.40), and a significant negative relationship with cognitive function was observed for age over 75 (OR: 0.10, 95% CI: 0.02–0.46, vs. <65 years). In Model 3, significant positive relationships with cognitive function were observed for cognitive SC (OR: 3.11, 95% CI: 1.43–6.78) and years of education (OR: 3.23, 95% CI: 1.48–7.05), and a significant negative relationship with the cognitive function was observed for age >75 (OR: 0.10, 95% CI: 0.02–0.50, vs. <65 years). In women, in Model 1, significant positive relationships with cognitive function were observed for structural SC (OR: 1.99, 95% CI: 1.14–3.47) and the years of education (OR: 3.00, 95% CI: 1.69–5.30), and significant negative relationship with the cognitive function was observed for age 65–74 (OR: 0.37, 95% CI: 0.18–0.74, vs. <65 years), and age >75 (OR: 0.24, 95% CI: 0.09–0.63, vs. <65 years). In Model 2, significant positive relationships with cognitive function were observed for the years of education (OR: 2.72, 95% CI: 1.55–4.78), and significant negative relationships with the cognitive function were observed for age 65–74 (OR: 0.38, 95% CI: 0.18–0.79, vs. <65 years), age >75 (OR: 0.22, 95% CI: 0.08–0.58, vs. <65 years) and BMI of 23.0–27.4 (OR: 0.89, 95% CI: 0.50–0.57). In Model 3, significant positive relationships with cognitive function were observed for structural SC (OR: 1.89, 95% CI: 1.08–3.31) and years of education (OR: 2.54, 95% CI: 1.41–4.58), and significant negative relationships with cognitive function were observed for age 65–74 (OR: 0.36, 95% CI: 0.17–0.75, vs. <65 years) and age >75 (OR: 0.23, 95% CI: 0.08–0.62, vs. <65 years).

## 4. Discussion

In this study, we assessed the relationship between SC and cognitive function among Japanese community-dwelling older adults by gender. Cognitive SC was associated with cognitive function in men, and structural SC was associated with cognitive function in women. There are several ways of analyzing the configuration of SC. There are cases in which SC is considered in the two configurations of cognitive SC and structural SC, and in other cases, it is considered in the two configurations of bonding and bridging SC. Cognitive SC is an individual’s sense of belonging to social groups, and structural SC is an individual’s participation in social groups. Bonding social capital refers to the strong ties with people in the same community that enable them to “get by”. Bridging social capital is the formal and informal links with other communities that enable people to “get ahead” [20]. A gender difference in the association between SC and health outcome was found in previous studies by using latter configuration: bonding SC was associated with self-rated health among men, and bridging SC was associated with self-rated health among women [25,26]. Although these are different ways of categorization, bonding SC is similar to cognitive SC in a way of building good relationships within homogenous environments, i.e., domestic neighborhoods. Furthermore, men are known to have more trust in others compared to women [27], which could explain why men could benefit from cognitive SC.

With respect to women, previous studies found that the number of group involvement was associated with self-rated health [25,26]. This finding is consistent with those of previous studies. Various social activities might have a positive impact on cognitive function by increasing psychosocial supports. We found no relationship between cognitive function and cognitive SC in women. This is likely because women tend to have many roles in their families, which result in fewer opportunities to bond in their neighborhoods.

This study could not assess reasons for gender differences in the relationship between SC and cognitive function. Gender differences may be due to mental processes, cultural background, and other factors that differ between men and women.

Our study had several limitations. First, this study included participants recruited from people undergoing medical examinations in two rural towns in Japan. Because the sample size was not large, the lower statistical power due to smaller sample size could have led to underestimations of any associations. These results have limits for the generalizability in rural areas. However, since 70% of Japan’s land area occupies the rural area, we believe that the result of this research is important for public health in rural areas. In the future, it is necessary to increase the target area and randomly sample the target population. Second, since our study had a cross-sectional design, we were unable to assess whether there was a causal relationship between cognitive function and SC. In order to clarify the causal relationship, it would be necessary to conduct a longitudinal study. Third, data used in our study were limited to the individual level SC. It is important to calculate neighborhood level SC and analyze the relationship with cognitive function.

## 5. Conclusions

In this study, we found the relationship between SC and cognitive function among Japanese community-dwelling older adults by gender. More specifically, cognitive function was associated with cognitive SC in men, and cognitive function was associated with structural SC in women. This suggests that it is important to approach dementia prevention differently with respect to gender. Further studies are needed to assess the effects of approaches to dementia prevention according to gender.

## Figures and Tables

**Table 1 ijerph-16-02142-t001:** Participant characteristics according to gender and cognitive level.

	Men				Women			
	Total	Cognitive Function Level		*p*-Value	Total	Cognitive Function Level		*p*-Value
		Low (CADi 0–7)	High (CADi 8–10)			Low (CADi 0–7)	High (CADi 8–10)	
Number of participants, *n*	187	76	111		304	130	174	
Cognitive SC score <5, %	61.9	50.7	69.5	0.01	51.4	52.5	50.6	0.75
Structural SC score ≥ 3, %	55.1	50.7	58.1	0.33	40.8	29.8	48.8	0.001
Age (years)								
<65, %	23.5	15.8	28.8	0.001	26.0	13.1	35.6	<0.001
65–74, %	66.3	64.5	67.6		59.9	65.4	55.7	
≥75, %	10.2	19.7	3.6		14.1	21.5	8.6	
BMI (kg/m^2^)								
<18.5 (underweight), %	4.8	0.0	8.1	0.06	5.9	6.9	5.2	0.71
18.5–22.9 (normal), %	49.2	53.9	45.9		50	46.9	52.3	
23.0–27.4 (overweight), %	39.0	40.8	37.8		36.2	36.9	35.6	
≥27.5 (obese), %	7.0	5.3	8.1		7.9	9.2	6.9	
Regular exercise, %	26.7	27.6	26.1	0.82	37.5	39.2	36.2	0.59
Depressive symptoms, %	38.0	47.3	31.8	0.03	38.4	45.7	32.9	0.02
Smoking, %	14.4	13.2	15.3	0.68	1.3	2.3	1.6	0.19
≥12 years of education, %	59.2	40.8	72.2	<0.001	52.7	32.0	68.0	<0.001

CADi: Cognitive Assessment of Dementia, iPad version; BMI: body mass index; SC: social capital.

**Table 2 ijerph-16-02142-t002:** Relationship between social capital and cognitive function according to gender.

Variable	Men						Women					
	Model 1		Model 2		Model 3		Model 1		Model 2		Model 3	
	OR	(95% CI)	OR	(95% CI)	OR	(95% CI)	OR	(95% CI)	OR	(95% CI)	OR	(95% CI)
Cognitive social capital												
Low	—	—	1.00	Reference	1.00	Reference	—	—	1.00	Reference	1.00	Reference
High	—	—	2.99	(1.42, 6.33)	3.11	(1.43, 6.78)	—	—	0.89	(0.53, 1.49)	0.93	(0.54, 1.59)
Structural social capital												
Low	1.00	Reference	—	—	1.00	Reference	1.00	Reference	—	—	1.00	Reference
High	1.01	(0.48, 2.09)	—	—	1.16	(0.53, 2.54)	1.99	(1.14, 3.47)	—	—	1.89	(1.08, 3.31)
Age (years)												
<65	1.00	Reference	1.00	Reference	1.00	Reference	1.00	Reference	1.00	Reference	1.00	Reference
65–74	0.69	(0.29, 1.67)	0.62	(0.24, 1.56)	0.62	(0.24, 1.60)	0.37	(0.18, 0.74)	0.38	(0.18, 0.79)	0.36	(0.17, 0.75)
≥75	0.19	(0.05, 0.78)	0.10	(0.02, 0.46)	0.10	(0.02, 0.50)	0.24	(0.09, 0.63)	0.22	(0.08, 0.58)	0.23	(0.08, 0.62)
BMI (kg/m^2^)												
<18.5 (underweight)	—	—	—	—	—	—	0.74	(0.25, 2.20)	0.76	(0.25, 2.28)	0.79	(0.26, 2.40)
18.5–22.9 (normal)	1.00	Reference	1.00	Reference	1.00	Reference	1.00	Reference	1.00	Reference	1.00	Reference
23.0–27.4 (overweight)	1.13	(0.54, 2.37)	1.31	(0.61, 2.84)	1.33	(0.59, 2.99)	1.09	(0.61, 1.96)	0.89	(0.50, 0.57)	1.04	(0.58, 1.89)
≥27.5 (obese)	0.98	(0.23, 4.12)	1.35	(0.32, 5.73)	1.25	(0.28, 5.69)	0.65	(0.24, 1.70)	0.60	(0.23, 1.59)	0.70	(0.26, 1.87)
Regular exercise												
No	1.00	Reference	1.00	Reference	1.00	Reference	1.00	Reference	1.00	Reference	1.00	Reference
Yes	1.02	(0.45, 2.29)	1.18	(0.51, 2.70)	1.25	(0.51, 3.06)	1.11	(0.63, 1.94)	0.96	(0.56, 1.66)	1.04	(0.59, 1.83)
Depressive symptoms												
No	1.00	Reference	1.00	Reference	1.00	Reference	1.00	Reference	1.00	Reference	1.00	Reference
Yes	0.42	(0.21, 0.86)	0.74	(0.35, 1.56)	0.61	(0.28, 1.34)	0.69	(0.39, 1.21)	0.59	(0.34, 1.03)	0.71	(0.40, 1.26)
Smoking												
Non-smoker	1.00	Reference	1.00	Reference	1.00	Reference	1.00	Reference	1.00	Reference	1.00	Reference
Smoker	1.17	(0.42, 3.28)	1.44	(0.50, 4.14)	1.43	(0.47, 4.41)	3.32	(0.25, 43.58)	2.54	(0.10, 65.55)	1.86	(0.08, 43.43)
Years of education												
<12	1.00	Reference	1.00	Reference	1.00	Reference	1.00	Reference	1.00	Reference	1.00	Reference
≥12	2.84	(1.37, 5.91)	3.05	(1.45, 6.40)	3.23	(1.48, 7.05)	3.00	(1.69, 5.30)	2.72	(1.55, 4.78)	2.54	(1.41, 4.58)

The dependent (outcome) variable is a high level of cognitive function, based on a CADi score ≥ 8. OR: odds ratio; CI: confidence interval.

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
