# Peer review of "Relationship between Individual Social Capital and Cognitive Function among Older Adults by Gender: A Cross-Sectional Study"

_ijerph, 2019, doi:10.3390/ijerph16122142_

Reviewer 1 Report

 The health consequence of social capital is a new topic in public health literature. The authors must provide a literature review (especially those conducted in Asia) either in the section of “Introduction” or in a new section before “Materials and Methods”.

The authors mentioned that “Previous studies [9] that examined the relationship between SC and health using these classifications have found no gender differences. As another study found that the factors related to cognitive function differ by gender [13], we hypothesized that there may be gender differences in the relationship between cognitive function and SC.” in Page 2. The authors must elaborate where this hypothesis comes from. Does the gender difference represent social inequality, as prior studies consistently report that social class is an important determinant of health.

The sample size is small and seems to be biased (187 men and 304 women, age ranged from 40 to 88 years). The authors must justify how this particular sample represents the Japanese rural society and how the findings of this study can be generalized in Japan.

Finally, the authors should provide a solid justification for the contribution of this article. 

Author Response

DearReviewer

Thank you for your comments to improve our manuscriptThe following is a point-by-point response to the questions and comments delivered in your letter.

Comment #1

 The health consequence of social capital is a new topic in public health literature. The authors must provide a literature review (especially those conducted in Asia) either in the section of “Introduction” or in a new section before “Materials and Methods”.

 Reply:

Thank you for your comments. We agree with your comments and have revised the Introduction.

< Revised > Introduction

For example, Kawachi et al. [10] found the association between social trust and total mortality in USA. Ferlander et al. [13] reported the relationship between SC and self-perceived health in Russia. In the Asia, Chuang et al. [14] observed a relationship between SC and drinking and smoking as lifestyle habits in Taiwan. Hamano et al. [11] and Nakamine et al. [12] found a relationship between SC and mental health in Japan. For the studies of cognitive function in Asia, Wang et al. [7] found a relationship between SC and mild cognitive impairment in China. Sakamoto et al. [8] reported a relationship between cognitive function and various social participation of community-dwelling older populations in Japan.

Comment #2

The authors mentioned that “Previous studies [9] that examined the relationship between SC and health using these classifications have found no gender differences. As another study found that the factors related to cognitive function differ by gender [13], we hypothesized that there may be gender differences in the relationship between cognitive function and SC.” in Page 2. The authors must elaborate where this hypothesis comes from. Does the gender difference represent social inequality, as prior studies consistently report that social class is an important determinant of health.

 Reply:

Thank you for your comments. We agree with your comments and have revised the Introduction.

< Revised > Introduction

For the cognitive function, the differences by gender have been clarified [15,16]. However, differences by gender for the SC have not been fully argued in the discussion on the association between SC and health. A recent study reported that differences by gender existed in social networks [17]. Moreover, a previous study reported that SC didn’t become an equally effective resource for everyone, and it gave an adverse effect on social equality between men and women [18]. Therefore, a better understanding of the function of SC is required to capture the association between SC and cognitive function by gender among rural older adults.

Comment #3 

The sample size is small and seems to be biased (187 men and 304 women, age ranged from 40 to 88 years). The authors must justify how this particular sample represents the Japanese rural society and how the findings of this study can be generalized in Japan.

 Reply:

Thank you for your comments. We agree with your comments and have revised the Discussion.

< Revised > Discussion

First, this study included participants recruited from people undergoing medical examinations in two rural towns in Japan. Because the sample size was not large, the lower statistical power due to smaller sample size could have led to underestimations of any associations. These results have limits for the generalizability in rural areas. However, since 70% of Japan's land area occupies the rural area, we think that the result of this research is important for public health in the rural area. In the future, it is necessary to increase the target area and randomly sample the target population.

Comment #4

Finally, the authors should provide a solid justification for the contribution of this article. 

 Reply:

Thank you for your comments. We agree with your comments and have revised the Author Contributions.

< Revised > Author Contributions

Author Contributions: Tomoko Ito, Kenta Okuyama and Takafumi Abe conceptualized and designed the study, performed the data analysis, and drafted the manuscript. Kenta Okuyama, Takafumi Abe and Miwako Takeda assisted with the implementation of data collection. Tsuyoshi Hamano and Kunihiko Nakano assisted with the conception and design of the study and revised the article critically for important intellectual content. Toru Nabika supervised the analysis and interpretation of the data, and helped draft and revise the manuscript. All authors have reviewed the manuscript and agreed with its contents.

Reviewer 2 Report

The authors examined the relationship individual social capital and cognitive function in Japanese adults living rural area. I felt that the work was of interest. However, I felt the divergence between title and the analyses methods. No suitable statistical analyses were found, e.g. relationship between male and female, or rural and urban area as described in the Title. The title should be changed, or Discussion should be improved/revised to be understood easily in the readers.

Author Response

DearReviewer

Thank you for your comments to improve our manuscriptThe following is a point-by-point response to the questions and comments delivered in your letter.

Comments and Suggestions for Authors

The authors examined the relationship individual social capital and cognitive function in Japanese adults living rural area. I felt that the work was of interest. However, I felt the divergence between title and the analyses methods. No suitable statistical analyses were found, e.g. relationship between male and female, or rural and urban area as described in the Title. The title should be changed, or Discussion should be improved/revised to be understood easily in the readers.

Reply:

Thank you for your comments. We agree with your comments and have revised the title, the purpose, Discussion and Conclusion.

< Revised > Title

Relationship between individual social capital and cognitive function among community-dwelling older adults by gender: a cross-sectional study

< Revised > Introduction (the purpose)

The aim of this study was to assess the relationship between individual cognitive and structured SC, and cognitive function among community-dwelling Japanese older adults by gender.

< Revised > Discussion and Conclusion

In this study, we found the relationship between SC and cognitive function among Japanese community-dwelling older adults by gender.

Round  2

Reviewer 1 Report

I think the authors have addressed my comments in general.

Reviewer 2 Report

None